# Unified framework for open quantum dynamics with memory

Felix Ivander[1], Lachlan P. Lindoy[2] & Joonho Lee [3,4] ✉

The dynamics of quantum systems coupled to baths are typically studied using the Nakajima-Zwanzig memory kernel ($\mathcal{K}$) or the influence functions ($\mathbf{I}$), particularly when memory effects are present. Despite their significance, formal connections between the two have not been explicitly known. We establish their connections by examining the system propagator for a $N$-level system linearly coupled to Gaussian baths with various types of system-bath coupling. For a certain class of problems, we devised a non-perturbative, diagrammatic approach to construct $\mathcal{K}$ from $\mathbf{I}$ for (driven) systems interacting with Gaussian baths, bypassing conventional projection-free dynamics inputs. Our work provides a way to interpret approximate path integral methods in terms of approximate memory kernels. Moreover, it offers a Hamiltonian learning procedure to extract the bath spectral density from reduced system trajectories, opening new avenues in quantum sensing and engineering. The insights we provide advance our understanding of non-Markovian dynamics and will serve as a stepping stone for future theoretical and experimental developments in this area.

Most existing quantum systems inevitably interact with the surrounding environment, often making a straightforward application of Schrödinger's equation impractical[1]. The main challenge in modeling these "open" quantum systems is the large Hilbert space dimension because the environment is much larger than the system of interest. Addressing this challenge is important in many disciplines, including solid state and condensed matter physics[2–4], chemical physics and quantum biology[5–8], quantum optics[9–12], and quantum information science[13–15]. In this work, we provide a unified framework for studying non-Markovian open quantum systems, which will help to facilitate a better understanding of open quantum dynamics and the development of numerical methods.

Various numerically exact methods have been developed to describe non-Markovian open quantum dynamics. Two of the most commonly used approaches are (1) the Feynman–Vernon influence functional path integral (INFPI)[16] based techniques, including the quasiadiabatic path-integral method of Makri and Makarov and its variants[17–26], hierarchical equations of motion (HEOM) methods[7,27,28], and time-evolving matrix product operator and related process tensor-based approaches[29–34] and (2) the Nakajima–Zwanzig generalized quantum master equation (GQME) techniques[1,35–37]. The INFPI formulation employs the influence functional ($\mathcal{I}$) that encodes the time-nonlocal influence of the baths on the system. In the GQME formalism, the analogous object to $\mathcal{I}$ is the memory kernel ($\mathcal{K}$), which describes the entire complexity of the bath influence on the reduced system dynamics. It is natural to intuit that $\mathcal{I}$ and $\mathcal{K}$ are closely connected and are presumably identical in their information content. Despite this, to the best of our knowledge, analytic and explicit relationships between the two have yet to be shown.

There have been several works that loosely connect these two frameworks. For instance, there is a body of work on numerically computing $\mathcal{K}$ with projection-free inputs using short-time system trajectories based on INFPI or other exact quantum dynamics methods[38–42]. The obtained $\mathcal{K}$ is then used to propagate system dynamics for longer times. Another line of work worth noting is the real-time path integral Monte Carlo algorithms for evaluating memory kernels exactly[43]. These works

[1]Quantum Science and Engineering, Harvard University, Cambridge, MA, USA. [2]National Physical Laboratory, Teddington TW11 0LW, United Kingdom. [3]Department of Chemistry and Chemical Biology, Harvard University, Cambridge, MA, USA. [4]Google Quantum AI, Venice, CA, USA. ✉ e-mail: joonholee@g.harvard.edu

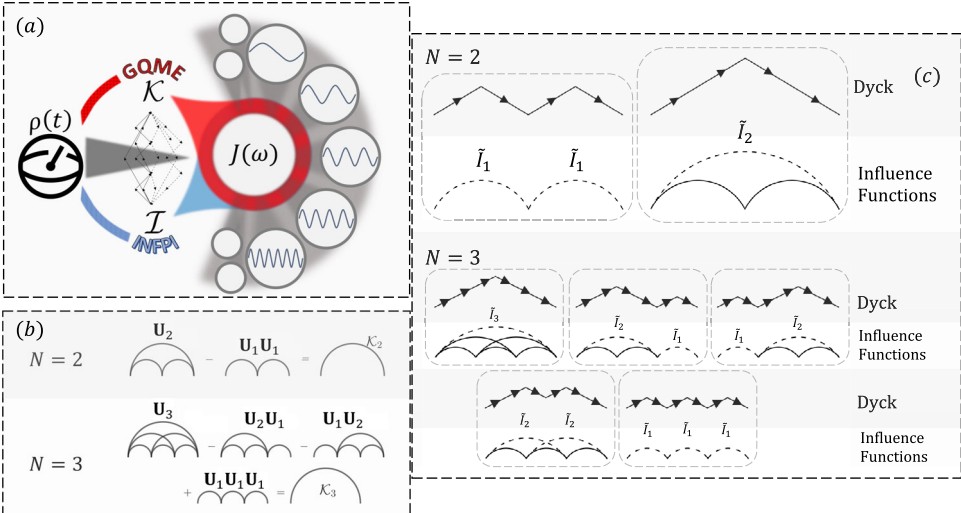

**Fig. 1 | Unification of open quantum dynamics framework for Class 1. a** An open quantum system, where the environment is characterized by the spectral density $J(\omega)$, can be described with the generalized quantum master equation (GQME) and the influence functional path integral (INFPI). The former distills environmental correlations through the memory kernels $\mathcal{K}$ while the latter through the influence functionals $\mathcal{I}$. In this work, we show both are related through Dyck Paths, and that, furthermore, we can use the Dyck construction for extracting $J(\omega)$ by simply knowing how the quantum system evolves. **b** Cumulant expansion of memory kernel. Examples through Eq. (6) for $N = 2$ and $N = 3$. Solid arcs of diameter $k$ filled with all possible arcs of diameters smaller than $k$ denote propagator $\mathbf{U}_k$. **c** Dyck path diagrams. Examples for $N = 2$ and $N = 3$ and their corresponding influence function diagrams, which composes $\mathcal{K}_2$ and $\mathcal{K}_3$, respectively. Solid lines denote influence functions $\mathbf{I}$ and dashed lines denote $\bar{\mathbf{I}}$.

took advantage of the real-time path integral approaches used to evaluate $\mathcal{I}$[44] to evaluate necessary matrix elements in computing the exact memory kernel. Nonetheless, they did not present any direct analytical relationship between the memory kernel and $\mathcal{I}$.

In this work, we present a unifying description of these non-Markovian quantum dynamics frameworks. In particular, we establish explicit analytic correspondence between $\mathcal{I}$ and $\mathcal{K}$. We present a visual schematic describing the main idea of our work in Fig. 1a. Readers interested in the relationship between our work and existing numerical tools are referred to Supplementary Note 3C.

## Results
### General setup
We consider a broad range of system-bath Hamiltonians in which the bath is Gaussian, and the system-bath Hamiltonian is bilinear. The total Hamiltonian is $\hat{H} = \hat{H}_S + \sum_j (\hat{H}_{B,j} + \sum_\alpha \hat{H}_{I,j,\alpha})$, with subscripts $j$ and $\alpha$ specifying the $j$th bath and the $\alpha$th interaction, respectively. While we do not limit the form of $\hat{H}_S$ in our discussion, we consider a quadratic (i.e., Gaussian) Hamiltonian for the baths, $\hat{H}_{B,j} = \sum_k \omega_{k,j} \hat{a}_{k,j}^\dagger \hat{a}_{k,j}$, where $\hat{a}_{k,j}$ can be fermionic or bosonic (it is also possible to treat baths consisting of noninteracting spins in a certain limit, see Supplementary Notes 3), and the bilinear interaction Hamiltonian, $\hat{H}_{I,j,\alpha} = \hat{S}_{j,\alpha} \otimes \hat{B}_{j,\alpha}$ with $\hat{S}_{j,\alpha}$ and $\hat{B}_{j,\alpha}$ being the system and bath operators, respectively. We also assume that the initial density matrix is separable between the system and each bath. There are four classes of problems that one may commonly encounter under the setup described:

1. *Class 1:* With only single $\alpha$ for all baths $j$ (such cases are henceforth indicated by dropping the subscript $\alpha$), $\{\hat{S}_j\}$ are all diagonalizable, and furthermore, that $\{\hat{S}_j\}$ are all simultaneously diagonalizable. That is, all terms in $\{\hat{H}_{I,j}\}$ commute. The spin-boson model, other models in the same universality class, and Frenkel exciton models for photosynthetic systems belong to this class.
2. *Class 2:* No terms in $\{\hat{s}_j\}$ commute but each term in $\{\hat{s}_j\}$ is diagonalizable. Generalizing the models in *Class 1* to multiple non-additive baths typically leads to this case. Such systems may arise when considering non-adiabatic dynamics of systems involving strong coupling of electronic degrees of freedom coupled to quantized photonic modes[32].

3. *Class 3:* There are common baths for some $\hat{H}_{I,j,\alpha}$ and $\{\hat{S}_{j,\alpha}\}$ may or may not commute. Examples of such baths arise when considering decoherence in models of coupled qubits[45].
4. *Class 4:* No terms in $\{\hat{S}_j\}$ commute and each term in $\{\hat{S}_j\}$ is not diagonalizable. The Anderson impurity model[46] is representative of this category.

We show in all three classes that one can relate $\mathcal{I}$ and $\mathcal{K}$ analytically. Furthermore, we show that one can obtain the bath spectral density from the reduced dynamics. Lastly, for *Class 1*, we show that a simple diagrammatic structure in the relationship between $\mathcal{I}$ and $\mathcal{K}$ can be found, which allows for efficient construction of $\mathcal{K}$ without approximations. We provide more details of *Class 1* in the main text, and additional details for other classes are available in the Supplementary Notes. Further, for *Class 1* models, we extend this analysis to consider driven systems, extending the analysis beyond the time-translationally invariant memory kernels observed for time-independent Hamiltonians.

### Path integral formulation
The time evolution of the full system is given by, $\rho_{\text{tot}}(t) = e^{-i\hat{H}t} \rho_{\text{tot}}(0) e^{i\hat{H}t}$. We discretize time and employ a Trotterized propagator,

$$e^{-i\hat{H}\Delta t} = e^{-i\hat{H}_S \Delta t/2} e^{-i\hat{H}_{\text{env}}\Delta t} e^{-i\hat{H}_S \Delta t/2} + O(\Delta t^3), \tag{1}$$

where $\hat{H}_{\text{env}} = \hat{H} - \hat{H}_S$. The initial total density matrix is assumed to be factorized into $\rho_{\text{tot}}(0) = \rho(0) \otimes (Z_j^{-1} \exp[-\beta_j \hat{H}_{B,j}])^{\otimes j}$ at inverse temperature $\beta_j$ where $Z_j = \text{Tr} \exp[-\beta \hat{H}_{B,j}]$. Then, one can show that the dynamics of the reduced system density matrix, $\rho(N\Delta t) = \rho_N = \text{Tr}_B [\rho_{\text{tot}}(N\Delta t)]$ (partial trace over all baths' degree of freedom), follows

$$\langle x_{2N}^+ | \rho_N | x_{2N}^- \rangle = \sum_{x_0^\pm \cdots x_{2N-1}^\pm} G_{x_0^\pm x_1^\pm} G_{x_1^\pm x_2^\pm} \ldots G_{x_{2N-1}^\pm x_{2N}^\pm}$$
$$\times \langle x_0^+ | \rho_0 | x_0^- \rangle \prod_\alpha \mathcal{I}_j(x_1^\pm, x_3^\pm, \cdots, x_{2N-1}^\pm), \tag{2}$$

where $G_{x_m^\pm x_{m+1}^\pm} = \langle x_m^+ | e^{-\frac{i\hat{H}_S \Delta t}{2}} | x_{m+1}^+ \rangle \langle x_{m+1}^- | e^{\frac{i\hat{H}_S \Delta t}{2}} | x_m^- \rangle$.

Restricting ourselves to problems in *Class 1* (details for other *Classes* are available in the Supplementary Notes), we consider $\hat{H}_I = \hat{S} \otimes \hat{B}$ where $\hat{S}$ is a system operator that is diagonal in the computational basis and $\hat{B} = \sum_k \lambda_k (\hat{a}_k^\dagger + \hat{a}_k)$ is a bath operator that is linear in the bath creation and annihilation operators (with the subscript $\alpha$ and $j$ dropped for clarity.) The discussion below can be applied to cases with multiple commuting $\hat{S} \otimes \hat{B}$ since $\mathcal{I}$ take simple product form, see Supplementary Note 1. We can show that the influence functional, $\mathcal{I}$, is pairwise separable,

$$
\begin{aligned}
\mathcal{I}(x_1^\pm, x_3^\pm, \cdots, x_{2N-1}^\pm) &= \prod_{n=1}^{N} I_{0,x_{2n-1}^\pm} \prod_{n=1}^{N-1} I_{1,x_{2n-1}^\pm x_{2n+1}^\pm} \\
&\times \prod_{n=2}^{N-1} I_{2,x_{2n-3}^\pm x_{2n+1}^\pm} \\
&\cdots \times I_{N-1,x_1^\pm x_{2N-1}^\pm}
\end{aligned}
\tag{3}
$$

where the *influence functions* $\mathbf{I}_k$ are defined in Supplementary Note 1, and are related to the bath spectral density, $J(\omega) = \pi \sum_k \lambda_k^2 \delta(\omega - \omega_k)$. For later use, we note that Eq. (2) can be simplified into

$$
\langle x_{2N}^+ | \rho_N | x_{2N}^- \rangle = \sum_{x_0^\pm} (\mathbf{U}_N)_{x_{2N}^\pm x_0^\pm} \langle x_0^+ | \rho_0 | x_0^- \rangle,
\tag{4}
$$

where $\mathbf{U}_N$ is the system propagator from $t = 0$ to $t = N\Delta t$. It is then straightforward to express $\mathbf{U}_N$ in terms of $\{\mathbf{I}_k\}$[19–21,42,47].

## The Nakajima–Zwanzig equation

The Nakajima–Zwanzig equation is a time-non-local formulation of the formally exact GQME. Assuming the time-independence of $\hat{H}_S$, the discretized homogeneous Nakajima–Zwanzig equation takes the form

$$
\rho_N = \mathbf{L} \rho_{N-1} + \Delta t^2 \sum_{m=1}^{N} \mathcal{K}_{N-m} \rho_{m-1},
\tag{5}
$$

where $\mathbf{L} \equiv (1 - \frac{i}{\hbar} \mathcal{L}_S \Delta t)$ with $\mathcal{L}_S \bullet \equiv [\hat{H}_S, \bullet]$ being the bare system Liouvillian and $\mathcal{K}_n$ is the discrete-time memory kernel at time step $n$. To relate $\mathcal{K}_N$ to $\{\mathbf{I}_k\}$, we inspect the reduced dynamics evolution operator $\mathbf{U}_N$ as defined in Eq. (4),

$$
\mathbf{U}_N = \mathbf{L} \mathbf{U}_{N-1} + \Delta t^2 \sum_{m=1}^{N} \mathcal{K}_{N-m} \mathbf{U}_{m-1}.
\tag{6}
$$

With this relation, one can obtain $\mathcal{K}_N$ from the reduced propagators $\{\mathbf{U}_k\}$. We observe setting $N = 1$ yields $\mathcal{K}_0 = \frac{1}{\Delta t^2}(\mathbf{U}_1 - \mathbf{L})$, since $\mathbf{U}_0$ is the identity. The memory kernel, $\mathcal{K}_0$, accounts for the deviation of the system dynamics from its pure dynamics (decoupled from the bath) within a time step. From setting $N = 2$, we get $\mathcal{K}_1 = \frac{1}{\Delta t^2}(\mathbf{U}_2 - \mathbf{U}_1 \mathbf{U}_1)$. This intuitively shows that $\mathcal{K}_1$ captures the effect of the bath that cannot be captured within $\mathcal{K}_0$. Similarly, for $N = 3$, $\mathcal{K}_2 = \frac{1}{\Delta t^2}(\mathbf{U}_3 - \mathbf{U}_2 \mathbf{U}_1 - \mathbf{U}_1 \mathbf{U}_2 + \mathbf{U}_1 \mathbf{U}_1 \mathbf{U}_1)$. This set of equations is similar to cumulant expansions, widely used in many-body physics and electronic structure theory[48,49]. Instead of dealing with higher-order $N$-body expectation values, we deal with higher-order $N$-time memory kernel in this context. The $N$-time memory kernel $\mathcal{K}_N$ is the $N$-th order cumulant in the cumulant expansion of the system operator. Unsurprisingly, these recursive relations lead to diagrammatic expansions commonly found in cumulant expansions[48], as shown in Fig. 1b.

## Relationship between $\mathcal{K}$ and I

Using this cumulant generation of $\mathcal{K}_N$ and by expressing $\{\mathbf{U}_k\}$ in terms of $\{\mathbf{I}_k\}$, we obtain a direct relationship between $\mathcal{K}_N$ and $\{\mathbf{I}_k\}_{k=0}^{k=N}$.

Specifically, we have

$$
\mathcal{K}_{0,ik} = \frac{1}{\Delta t^2} \left[ \sum_j G_{ij} I_{0,j} G_{jk} - L_{ik} \right]
\tag{7}
$$

$$
\mathcal{K}_{1,im} = \frac{1}{\Delta t^2} \sum_{jk} G_{ij} I_{0,j} F_{jk} \tilde{I}_{1,jk} I_{0,k} G_{km}
\tag{8}
$$

$$
\begin{aligned}
\mathcal{K}_{2,ip} = \frac{1}{\Delta t^2} \sum_{jkn} G_{ij} F_{jk} F_{kn} \Big( & \tilde{I}_{2,jn} I_{1,jk} I_{1,kn} \\
& + \tilde{I}_{1,jk} \tilde{I}_{1,kn} \Big) I_{0,j} I_{0,k} I_{0,n} G_{np}
\end{aligned}
\tag{9}
$$

$$
\begin{aligned}
\mathcal{K}_{3,il} = \frac{1}{\Delta t^2} \sum_{jknp} & G_{ij} F_{jk} F_{kn} F_{np} I_{0,j} I_{0,k} I_{0,n} I_{0,p} G_{pl} \\
& \Big\{ \tilde{I}_{3,jp} I_{2,jn} I_{2,kp} I_{1,jk} I_{1,kn} I_{1,np} \\
& + I_{1,kn} \big( \tilde{I}_{2,jn} \tilde{I}_{2,kp} I_{1,jk} I_{1,np} + \tilde{I}_{2,kp} \tilde{I}_{1,jk} I_{1,np} \\
& + \tilde{I}_{2,jn} \tilde{I}_{1,np} I_{1,jk} \big) + \tilde{I}_{1,jk} \tilde{I}_{1,kn} I_{1,np} \Big\} \\
& \vdots
\end{aligned}
\tag{10}
$$

where we define $\mathbf{F} = \mathbf{GG}$ (bold-face for denoting matrices) and $\tilde{I}_{k,ij} = I_{k,ij} - 1$. We emphasize that Eqs. (7) to (10) are exact up to the Trotter discretization error and valid for any coupling strengths in the models considered in this work. By definition, earlier $\mathcal{K}_N$ contains shorter memory effects and will thus appear simpler.

This series of equations is a part of the main result of this work, showing explicitly how $\mathcal{K}_N$ is diagrammatically constructed in terms of influence functions from $\mathbf{I}_0$ to $\mathbf{I}_N$. This construction can easily show the computational effort of computing $\mathcal{K}_N$. We sum over an additional time index for each time step. This gives a computational cost that scales exponentially in time, $\mathcal{O}(N_{\text{dim}}^{2N})$ where $N_{\text{dim}}$ is the dimension of the system Hilbert space. In Supplementary Note 3E, we present further details on the general algorithm for calculating higher-order memory kernels, exploiting a non-trivial diagrammatic structure to express them in terms of $\mathbf{I}$ and $\tilde{\mathbf{I}}$.

It can be inferred from Eqs. (8) to (10) that each term in $\mathcal{K}_N$ is represented uniquely by each Dyck path[50–52] of order $N$. Hence, one can construct $\mathcal{K}_N$ by generating the respective set of Dyck paths and associating each path with a tensor contraction of influence functions. This is illustrated in Fig. 1c and further detailed in Supplementary Note 3E. This observation reveals some new properties of $\mathcal{K}_N$. First, the number of terms in $\mathcal{K}_N$ is given by the $N$-th Catalan's number[51,52] $C_N = \frac{1}{N+1} \binom{2N}{N}$ (i.e., $\mathcal{K}_4$ has 14 such terms, $\mathcal{K}_5$ has 42, then 132, 429, 1430, 4862, 16796, 58786, ...). We note that Catalan's number appeared in ref. 47 when analyzing an approximate numerical INFPI method. See Supplementary Note 3E for more information.

Scrutinizing the relationship of $\mathcal{K}$ and $\mathbf{I}$, presented in Supplementary Note 3E, further, we can observe how $\mathcal{K}$ decays asymptotically. As is well-known, for typical condensed phase systems $I_{k,ij} \to 1$ for $k \to \infty$[17,53]. Similarly, because $\tilde{I}_{k,ij} \ll 1$ for large $k$, those terms with larger multiplicities contribute less to $\mathcal{K}_N$ and decay exponentially to zero as multiplicity grows. In fact, for condensed phase systems, the decay of $\mathbf{I}_N$ and $\mathcal{K}_N$ is often rapid, which motivated the development of approximate INFPI methods[17–20,53] and other approximate GQME methods[37,54–56].

With our new insight, approximate INFPI methods can be viewed through the lens of the corresponding memory kernel content (and vice versa). As an example, we shall discuss the iterative quasiadiabatic

path-integral methods[17,18,53]. In these methods, $I_{k,ij}$ is set to unity beyond a preset truncation length $k_{max}$. For simplicity, let us consider $k_{max} = 1$, and hence $I_{k,ij} = 1$ and $\tilde{I}_{k,ij} = 0$ for $k > k_{max}$. We now inspect what this approximation entails for $\mathcal{K}_N$. First, no approximation is applied to $\mathcal{K}_0$ and $\mathcal{K}_1$. Then, in $\mathcal{K}_2$ (Eq. (9)),

$$(\tilde{I}_{2,jn} I_{1,jk} I_{1,kn} + \tilde{I}_{1,jk} \tilde{I}_{1,kn}) \rightarrow \tilde{I}_{1,jk} \tilde{I}_{1,kn}. \tag{11}$$

Similarly, in $\mathcal{K}_3$ (Eq. (10)), the only surviving contribution is from $\tilde{I}_{1,jk} \tilde{I}_{1,kn} \tilde{I}_{1,np}$. We hope such a direct connection between approximate methods will inspire the development of more efficient and accurate methods.

The time-translational structure of the INFPI formulation and its Dyck-diagrammatic structure allow for a recursive deduction of $\mathbf{I}_N$ from $\mathcal{K}_N$, which is the inverse map of Eqs. (8) to (10). We first observe that

$$\mathbf{I}_0 = \mathbf{G}^{-1}(\delta t^2 \mathcal{K}_0 + \mathbf{L})\mathbf{G}^{-1} \tag{12}$$

where we obtained $\mathbf{I}_0$ from $\mathcal{K}_0$. One can then show that

$$I_{1,jk} = 1 + \Delta t^2 \frac{(\mathbf{G}^{-1}\mathcal{K}_1\mathbf{G}^{-1})_{jk}}{F_{jk} I_{0,j} I_{0,k}}. \tag{13}$$

using $\mathcal{K}_1$ and $\mathbf{I}_0$. Similarly, inspecting the expression for $\mathcal{K}_2$ gives us

$$I_{2,jn} = 1 + \frac{\left[\Delta t^2(\mathbf{G}^{-1}\mathcal{K}_2\mathbf{G}^{-1})_{jn} - \sum_k F_{jk} F_{kn} \tilde{I}_{1,jk} \tilde{I}_{1,kn} I_{0,j} I_{0,k} I_{0,n}\right]}{\sum_k F_{jk} F_{kn} I_{1,jk} I_{1,kn} I_{0,j} I_{0,k} I_{0,n}}, \tag{14}$$

where $\tilde{I}_{1,jk} = I_{1,jk} - 1$ as well as $I_{0,i}$ are obtained from the previous two relations.

## Spectral density learning

In Supplementary Note 3F, we present a general recursive procedure using the Dyck paths and how to obtain the bath spectral density from $I_k$. As a result, we achieve the following mapping from left to right,

$$\rho \rightarrow \mathbf{U} \rightarrow \mathcal{K} \rightarrow \mathbf{I} \rightarrow J(\omega). \tag{15}$$

A remarkable outcome of this analysis is that one can completely characterize the environment (i.e., $J(\omega)$), by inspecting the reduced system dynamics. Such a tool is powerful in engineering quantum systems in experiments where we have access to only the reduced system Hamiltonian and reduced system dynamics, but lack information about the environment. Furthermore, this approach provides an alternative to quantum noise spectroscopy[57,58]. This type of Hamiltonian learning with access only to subsystem observables has been achieved for other simpler Hamiltonians[59,60]. To our knowledge, our work is the first to show this inverse map for the Hamiltonian considered here.

Note that the expression Eq. (13) can become ill-defined when $\mathbf{F}$ is diagonal. This occurs when $\hat{H}_S$ is diagonal and commutes with $\hat{H}_{env}$, constituting a purely dephasing dynamics. In that case, the reduced system dynamics is governed only by the diagonal elements of $\mathbf{I}$. Similarly, $\mathcal{K}$ is diagonal, as clearly seen in our Dyck path construction. As a result, the map $\mathcal{K} \leftrightarrow \mathbf{I}$ is no longer bijective in that we cannot obtain off-diagonal elements of $\mathbf{I}$. Regardless, one can still extract $J(\omega)$ using only the diagonal elements of $\mathbf{I}$ via inverse cosine transform. One may worry Eq. (14) could also become ill-conditioned when its denominator vanishes, but $\hat{H}_S$ is not diagonal. If that were the case, the propagator $\mathbf{U}_2$ would become zero. Therefore, this condition cannot be satisfied in general. Finally, we remark that generalization to extract the $\mathcal{I}_\alpha$ of multiple baths through a single central system is possible and straightforward. See Supplementary Note 3F for more details.

## Generalization to driven systems

While analysis up to this point considered general time-independent systems, in many scenarios, e.g., of biological or engineering relevance, particularly for quantum control applications[61], a time-dependent description of the system is necessary. In such cases, $\mathcal{K}$ loses its time-translational properties and should depend on two times. Consequently, Eq. (6) cannot be applied. To overcome this, we factorize $\mathcal{K}_{N+s,s}$ into time-dependent and time-independent parts. This can be achieved straightforwardly, as follows: one observes upon the inclusion of time-dependence in $\hat{H}_S$, the terms that are affected in $\mathcal{K}_N$, Eqs. (7) to (10), are only the bare system propagators $\mathbf{G}$ and $\mathbf{F}$. We define the remainder as tensors with $N$ number of indices, $T_{N;x_{s+2},x_{s+4},\ldots,x_{s+2N}}$, which includes all the influence of the bath between $N$-time steps. These tensors only need to be computed once and reused for a later time. Then, one builds the kernels via tensor contraction over two tensors,

$$\mathcal{K}_{N+s,s;x_{s+2N+2},x_s} = \frac{1}{\Delta t^2} \sum_\bullet P^{N+1+s,s}_{x_s,\bullet,x_{s+2N+2}} T_{N;\bullet}, \tag{16}$$

where $\bullet$ denotes indices, $x_{s+2}, \ldots, x_{s+2N}$, and the tensor $P^{N+1+s,s}_{x_s,\bullet,x_{2N+s}}$ encapsulates the time-dependence of the system Hamiltonian and is constructed only out of bare system propagators. The tensor, $T_{N;\bullet}$, then consists only of influence functions, up to $\mathbf{I}_N$. The construction of these tensors is straightforward with $T_{N;\bullet}$ following the Dyck path construction presented for time-independent system dynamics. On the surface, the $T_{N;\bullet}$ tensor appears to be related to the process tensor[33,34]: $\mathbf{T}$ represents $\mathbf{K}$ upon the contraction with $\mathbf{P}$, but the process tensor is used to construct $\mathbf{U}$ when contracted with $\mathbf{P}$. Subsequently, there is a non-trivial rearrangement of the terms to write $\mathbf{K}$ in terms of the process tensor. The simple relationship between $\mathbf{T}$ and $\mathbf{K}$ in Eq. (16) is our unique contribution. More detailed analysis and relevant numerical results for open, driven system dynamics are presented in Supplementary Note 3H.

## Numerical verification

While the discussion above applies to a generic system linearly coupled to a Gaussian bath (or multiple such baths if they couple additively), we discuss the spin-boson model for further illustration. The spin-boson model is an archetypal model for studying open quantum systems[62]. The model comprises a two-level system coupled linearly to a bath of harmonic oscillators. Hence, it and its generalizations have been used to understand various quantum phenomena: transport, chemical reactions, diode effect, and phase transitions[63].

We use $\hat{H}_S = \epsilon \sigma_z + \Delta \sigma_x$, coupled via $\sigma_z$ to a harmonic bath with spectral density ($\omega \geq 0$)[62]

$$J(\omega) = \pi \sum_k \lambda_k^2 \delta(\omega - \omega_k) = \frac{\xi \pi}{2} \frac{\omega^s}{\omega_c^{s-1}} e^{-\omega/\omega_c}, \tag{17}$$

where $J(-\omega) = -J(\omega)$, $\xi$ is the Kondo parameter, and $s$ is the Ohmicity. All reference calculations were performed using the HEOM method[28,64,65]. Details of the HEOM implementation used here are provided in Supplementary Note 7.

In Fig. 2, we investigate a series of spin-boson models corresponding to weak and intermediate coupling to an Ohmic environment ($s = 1$) as well as strong coupling to a subohmic environment ($s = 0.5$). In panels (a, b), we observe that the decay of $\tilde{\mathbf{I}}_N$ is rapid for the Ohmic cases. This translates to a similarly rapid decay for the respective $\mathcal{K}_N$, although one can see that both $\tilde{I}_N$ and $\mathcal{K}_N$ are overall scaled larger in the strong coupling regime. This is to be contrasted with the results for the strongly coupled subohmic environment shown in panel (c). The decay of the $\tilde{\mathbf{I}}_N$ is slow, accompanied by a similarly slow decay of $\mathcal{K}_N$. Interestingly, the rates by which both $\tilde{\mathbf{I}}_N$ and $\mathcal{K}_N$ decay are similar, which we observe to be exponential. We also see

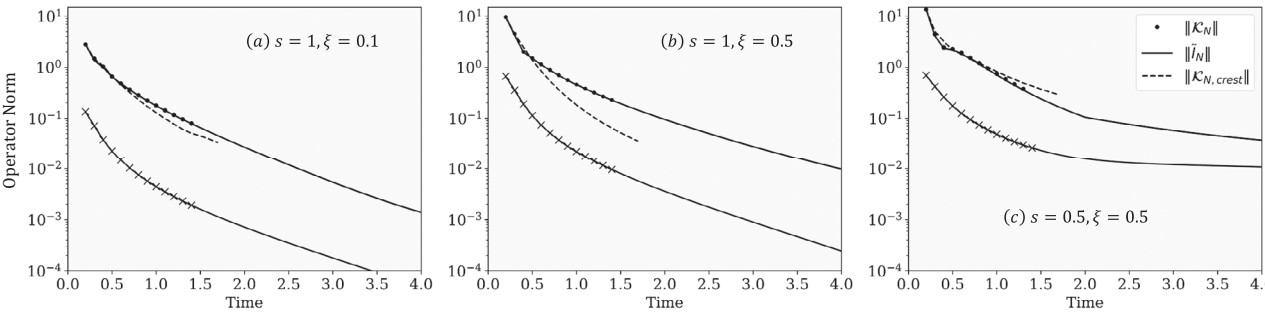

**Fig. 2 | Numerical verification of the Dyck construction.** Operator norm of $\bar{\mathbf{I}}_N$ (Light) and $\mathcal{K}_N$ (Dark) as a function of $N\Delta t$. Lines denote $\bar{\mathbf{I}}_N$ computed from analytic expressions and $\mathcal{K}_N$ from post-processing exact numerical results via the transfer tensor method[40]. Circles denote $\mathcal{K}_N$ from the Dyck diagrammatic method, and crosses are $\bar{\mathbf{I}}_N$ obtained via the inverse map discussed in Eqs. (13) and (14). Dashed lines denote the operator norm of the crest term of $\mathcal{K}_N$ (the Dyck path diagram with the highest height). Parameters used are: $\Delta = 1$ (other parameters are expressed relative to $\Delta$), $\epsilon = 0$, $\beta = 5$, $\Delta t = 0.1$, $\omega_c = 7.5$, and $\xi = 0.1$ and $s = 1$ (**a**), $\xi = 0.5$ and $s = 1$ (**b**), and $\xi = 0.5$ and $s = 0.5$ (**c**).

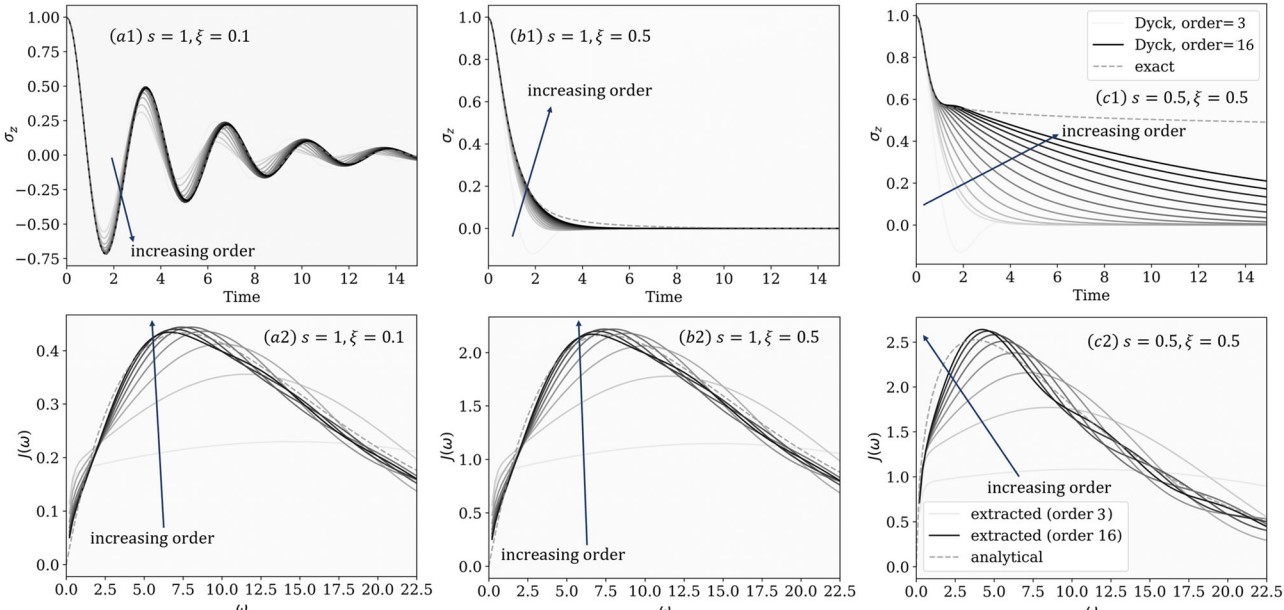

**Fig. 3 | Dynamics of spin-boson model with truncated Dyck paths. a**1, **b**1, **c**1 Magnetization ($\langle\sigma_z(t)\rangle$) dynamics predicted using $\mathcal{K}$ constructed via Dyck diagrams with increasing truncation orders (from light to darker colors) compared to exact results (see Supplementary Note 6). **a**2, **b**2, **c**2 Bath spectral densities extracted through the Dyck diagrammatic method with increasing truncation order (from white to black colors) compared to exact spectral densities (dashed), see Supplementary Note 3F for more details. These results come from numerically exact trajectories, initiated from linearly independent initial states $\rho_1(0) = \frac{1}{2}(\mathbf{1} + \sigma_z), \rho_2(0) = \frac{1}{2}(\mathbf{1} - \sigma_z), \rho_3(0) = \frac{1}{2}(\mathbf{1} + \sigma_x), \rho_4(0) = \frac{1}{2}(\mathbf{1} + \sigma_x + \sigma_y + \sigma_z)$. Parameters used are: $\Delta = 1$ (other parameters are expressed relative to $\Delta$), $\epsilon = 0$, $\beta = 5$, $\Delta t = 0.1$ (**a**1, **b**1, **c**1) or $\Delta t = 0.05$ (**a**2, **b**2, **c**2), $\omega_c = 7.5$, and $\xi = 0.1$ and $s = 1$ (**a**1 and **a**2), $\xi = 0.5$ and $s = 1$ (**b**1 and **b**2), or $\xi = 0.5$ and $s = 0.5$ (**c**1 and **c**2).

perfect agreement between $\mathcal{K}_N$ constructed from our Dyck diagrammatic method and those obtained by numerically post-processing exact trajectories via the transfer tensor method[40]. Lastly, we construct $\bar{\mathbf{I}}_N$ from $\mathcal{K}_N$ up to $N = 16$ as exemplified in Eqs. (13) and (14) and observe perfect agreement between our $\bar{\mathbf{I}}_N$ and those computed from its known analytic formula.

We note that the term with $\bar{\mathbf{I}}_N$ (multiplicity of 1) contributes the most to the memory kernel, $\mathcal{K}_N$ for all parameters considered in our work. We refer to this term as the "crest" term, which corresponds to the Dyck path that goes straight to the top and down straight to the bottom, having the tallest height. We see a small difference between the crest term norm and the full memory kernel norm in Fig. 2, indicating that the memory kernel is dominated by the crest term. Since the decay of $\bar{\mathbf{I}}_N$ is directly related to the decay of the bath correlation function, one can also make connections between the memory kernel decay and the bath correlation function decay. Nonetheless, for a stronger system-bath coupling (e.g., Fig. 2b) and for cases with a long-lived memory (e.g., Fig. 2c), terms other than the crest term contribute non-negligibly, making general analysis of the memory kernel decay challenging.

The cost to numerically compute $\mathcal{K}_N$ scales exponentially with $N$. Nevertheless, it is possible to exploit the decay of $\bar{\mathbf{I}}_N$, which is rapid for some environments, e.g., ohmic baths, in turn signifying the decay behavior of $\mathcal{K}_N$. This allows truncating the summation in Eq. (5), enabling dynamical propagation to long times (with linear costs in time) as usually done in small matrix path integral methods[19,20] and GQME[40] methods. We show in panels (a1) and (b1) of Fig. 3 that this procedure applied to a problem with a rapidly decaying $\mathcal{K}_N$ quickly converges to the exact value with a reasonably low-order. On the other hand, for environments with slowly decaying $\bar{\mathbf{I}}_N$, the truncation scheme struggles to work effectively. For a strongly coupled subohmic environment, as shown in Fig. 3c1, one would need truncation orders

beyond the current computational capabilities of our implementation (about 16) to converge to the exact value. Nonetheless, this illustrates that our direct construction of $\mathcal{K}_N$ can recover exact dynamics if sufficiently high-order is used. Furthermore, the construction is non-perturbative and can be applied to strong coupling problems. We note that describing quantum phase transitions at $T = 0$ would require capturing the algebraic decay in $\mathbf{I}_N$[29]. Our analysis can, in principle, capture such a slow decay as our approach is exact but will require further optimization in the underlying numerical algorithms for practical applications.

Finally, in Fig. 3a2, b2, c2, we show the extraction of spectral densities $J(\omega)$ for three distinct environments. The extracted $J(\omega)$ converges to the analytical value as we obtain the influence functions to higher orders. This shows that we can indeed invert the reduced system dynamics to obtain $J(\omega)$ given the knowledge of the system Hamiltonian, which ultimately characterizes the entire system-bath Hamiltonian. Nonetheless, the accuracy of the resulting $J(\omega)$ depends on the highest order of $\mathbf{I}_k$ we can numerically extract. The cost of extracting $\mathbf{I}_k$ scales exponentially in $k$ without approximations, so there is naturally a limit to the precision of $J(\omega)$ in practice. Furthermore, we show how this procedure can extract highly structured spectral densities as well in Supplementary Note 8 and Supplementary Fig. 9. New opportunities await in using approximately inverted $\mathbf{I}_k$ and quantifying the error in the resulting $J(\omega)$.

## Discussion

In this work, we provide analytical analysis along with numerical results that show complete equivalence between the memory kernel ($\mathcal{K}$) in the GQME formalism and the influence function ($\mathbf{I}$) used in INFPI. Our analysis applies to a broad class of general (driven) systems interacting bilinearly with Gaussian baths. Furthermore, we showed that one can extract the bath spectral density from the reduced system dynamics with the knowledge of the reduced system Hamiltonian $\hat{H}_S$. We believe that this unified framework for studying non-Markovian dynamics will facilitate the development of new analytical and numerical methods that combine the strengths of both GQME and INFPI. For example, deep connections between the present work and recent matrix product state (MPS)-based approaches invite ideas that would efficiently extract the environmental spectral density from reduced system dynamics[29,31–34].

## Methods

Details pertaining to analytical derivation of results in this work, as well as numerical implementations, are provided in the Supplementary Notes.

## Data availability

Data generated in this study is available on GitHub (https://github.com/JoonhoLee-Group/Unified_Framework_OQ_Code_and_Data) and Zenodo at ref. 66.

## Code availability

Simulation codes used in this study are available on GitHub (https://github.com/JoonhoLee-Group/Unified_Framework_OQ_Code_and_Data) and Zenodo at ref. 66.

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

## Acknowledgements

F.I. and J.L. were supported by Harvard University's startup funds. L.P.L. acknowledges the support of the Engineering and Physical Sciences Research Council [grant EP/Y005090/1]. We thank Nathan Ng, David Reichman, Dvira Segal, and Jonathan Keeling for stimulating discussions, Tom O'Brien for discussions on Hamiltonian learning, and Hieu Dinh for providing a code to generate the Dyck path. Computations were carried out partly on the FASRC cluster supported by the FAS Division of Science Research Computing Group at Harvard University. This work also used the Delta system at the National Center for Supercomputing Applications through allocation CHE230078 from the Advanced Cyberinfrastructure Coordination Ecosystem: Services & Support (ACCESS) program, which is supported by National Science Foundation grants #2138259, #2138286, #2138307, #2137603, and #2138296.

## Author contributions

F.I., L.P.L., and J.L. contributed to the conception, execution, analysis, and writing of the work.

## Competing interests

The authors declare no competing interests.
