## [Peer Review File · Nature Communications]

Unified Framework for Open Quantum Dynamics with MemoryREVIEWER COMMENTS

Reviewer #1 (Remarks to the Author):

The manuscript presents a direct and nonperturbative connection between the influence functional and the memory-kernel master equation for a generic open quantum system interacting with a Gaussian (bosonic) bath – refer to equations (7)-(10) (and equation (16) in the driven scenario) for the derivation of the memory kernel from the influence functional, and equations (12)-(14) for the inverse procedure. This connection enables the authors to propose a general methodology, offering an alternative approach to conventional quantum noise spectroscopy, for characterizing the spectral density of the environment by knowing the open-system dynamics only. Additionally, the manuscript investigates the spin-boson model with an Ohmic-like spectral density to exemplify both the insights gained from the direct link between the influence functional and the memory kernel and the reconstruction of the spectral density.

The manuscript presents an interesting result, i.e., the direct connection between the memory kernel and the influence functional, that is novel, clearly described and merits publication. However, in order to evaluate its conceptual relevance and potential impact more comprehensively, further analysis is needed.

More explicitly, as indeed acknowledged by the authors themselves, in the literature there are already well-established connections between memory-kernel master equations and the propagators of the dynamics that, as the approach proposed here, are nonperturbative and related with the path-integral formulation of the reduced dynamics, see for example the approach proposed in [38,39] or the transfer tensor method (TTM) [40]. Even more broadly speaking, while approaches like QUAPI do not explicitly address the memory-kernel master equation, they do facilitate the linking of open-system dynamics with the influence functional, enabling, for instance, the reconstruction of environmental spectral density from reduced dynamics.

What I think is missing is a thorough comparison, say with TTM and QUAPI (but also other approaches would work if they were more suitable) to clarify the relevance and novelty of what is presented here. Specifically, I would first consider a simple – even exactly solvable – model to exemplify more explicitly the differences between the relations defining the

distinct approaches and the insights that can be gained through them. Subsequently, I would utilize a more structured scenario to compare the complexity and efficiency of the methodologies.

Reviewer #2 (Remarks to the Author):

Ivander, Lindoy and Lee present a framework for connecting the generalised quantum master equation (GQME) to influence function methods. Their method is elegant, and as an example they show it is effective for simulating the spin-boson model. The paper is well written and the method very clearly explained with appropriate diagrams.

As the method presented seems closely related to other methods - which I will ask about further below - I am not sure whether this paper merits publication in Nature Communications. However, it is an interesting paper for specialists in the field - and the question is whether it represents a sufficiently important advance to fit within the journal's scope.

My main question is: can the authors say how their approach is different from Makri's small matrix path integral approach, which in turn is similar to Cerrillo and Cao's transfer tensor method? There is some mention of this in appendix A, but could I ask the authors to expand on this? What does their technique enable that these other methods do not?

When I read in the introduction to the paper that "efficient and completely general" descriptions of open quantum systems are yet to be found, I thought this paper would be presenting such a method. Do the authors believe this to be the case? If so, I think they would need to move beyond linear coupling to a harmonic bath. Indeed, is it possible to use the technique, for example, for fermionic baths? If so, I would ask the authors to provide an example. If this is not the case, then I would ask the authors to reword the introduction to make this clear.

In my view the most important part of this paper is the observation that one can reconstruct the spectral density $J(\omega)$ from the time evolution of ρ - and this is nicely illustrated

with a couple of examples. How general is this observation? Does it only apply for linearly coupled harmonic baths or more generally? The authors should explicitly say in the paper where this results holds.

Now on to some more specific comments:

1. At the bottom of col 1, page 4. Can the authors comment on the connecting between the T tensor and the process tensor as introduced in ref 31 of the paper? Also connect the PT to Eq. A71 and A72.
2. Fig 3, for the spin boson model, can the method capture the phase transition in this model?
3. More generally, how do computation times/resources using this method compare to other techniques (like TEMPO)?
4. Eq. A28. Is there a K here that should be an F?
5. Can the authors clarify the comment below Eq. A60 about frequencies that make the denominator go to zero? I could not quite follow the comment about these not contributing to $F(\omega)$, sorry! I can see that $\delta k > 0$ are H.c. of $\delta k < 0$ but does this not leave a real part?
5. In a comment in the caption of Fig 11, the author say that going from K->I for high orders is infeasible. Why? Does this limit the applicability of the method?
6. A further question: how well can one extract U from experimental measurements? How many such measurements need to be done to do this to a particular accuracy?

Reviewer #3 (Remarks to the Author):

The article discusses the dynamics of an open quantum system, described either using the Nakajima-Zwanzig memory kernel or the Feynman-Vernon influence functional path integral. The main result is a formal, diagrammatic connection between the two approaches, which allows for the expression of the memory kernel through the influence functions (or vice versa), using the reduced system evolution operator to link the two methods. As an application of this connection, the authors show how to construct the bath spectral density using only the reduced dynamics as input.

Altogether, I find that the results are interesting and clearly contribute to the field. I therefore recommend publication. However, I have some minor remarks:

- The authors mention that their construction of K using I is non-perturbative. This is not clear to me. In the context of GQME approaches one often expands the memory kernel K in orders of the system-environment coupling. When this is done in time-space one finds that higher-order memory kernel terms contribute "later in time" than lower order ones (in the sense of first non-vanishing Taylor coefficients). In this sense, e.g. K_3 from the paper contains higher order contributions than K_2 , which seems to give a perturbative character to the set of equations (7)-(10).

- The article is concerned with systems with bosonic environments. However, both the Feynman-Vernon path integral approach and the Nakajima-Zwanzig approach are also commonly used to also describe fermionic environments, e.g. in the description of quantum transport. Are there principle limitations in finding a similar formal connection between I and K also in the fermionic case?

Summary of Changes Made

We have made the following changes in response to queries by the referees:

1. We fixed several typos and clarified the results as pointed out by the referees.
2. We updated the reference list with recently published preprints and papers relevant to the referees' suggestions.
3. We have reorganized and expanded the Appendix to clarify our contributions and provide details on our results' generality.

Conventions used in this document

Responses to referee reports appear in black.

Additions / changes to the manuscript appear in blue.

Original referee reports appear in quoted black.

Response to Referee 1:

Referee Comment 0:

The manuscript presents a direct and nonperturbative connection between the influence functional and the memory-kernel master equation for a generic open quantum system interacting with a Gaussian (bosonic) bath – refer to equations (7)-(10) (and equation (16) in the driven scenario) for the derivation of the memory kernel from the influence functional, and equations (12)-(14) for the inverse procedure. This connection enables the authors to propose a general methodology, offering an alternative approach to conventional quantum noise spectroscopy, for characterizing the spectral density of the environment by knowing the open-system dynamics only. Additionally, the manuscript investigates the spin-boson model with an Ohmic-like spectral density to exemplify both the insights gained from the direct link between the influence functional and the memory kernel and the reconstruction of the spectral density.

The manuscript presents an interesting result, i.e., the direct connection between the memory kernel and the influence functional, that is novel, clearly described and merits publication. However, in order to evaluate its conceptual relevance and potential impact more comprehensively, further analysis is needed.

Response:

We thank the referee for reading our work, for providing helpful feedback, and for finding that it merits publication.

Referee Comment 1:

More explicitly, as indeed acknowledged by the authors themselves, in the literature there are already well-established connections between memory-kernel master equations and the propagators of the dynamics that, as the approach proposed here, are nonperturbative and related to the path-integral formulation of the reduced dynamics, see for example the approach proposed in [38,39] or the transfer tensor method (TTM) [40]. Even more broadly speaking, while approaches like QUAPI do not explicitly address the memory-kernel master equation, they do facilitate the linking of open-system dynamics with the influence functional, enabling, for instance, the reconstruction of environmental spectral density from reduced dynamics. What I think is missing is a

thorough comparison, say with TTM and QUAPI (but also other approaches would work if they were more suitable) to clarify the relevance and novelty of what is presented here.

Response:

We thank the referee for their comment. In Appendix A of our original submission, we made brief remarks on the relationship of our approach with TTM and SMatPI (identical to QUAPI when performed exactly). We used this opportunity to expand our discussion further. There are three points we would like to emphasize:

1. Our manuscript is not about developing efficient numerical techniques for non-Markovian dynamics. Instead, we offer new analytical insights that connect two major paradigms in understanding non-Markovian dynamics: influence functional-based path integral (e.g., QUAPI, SMatPI) methods and memory kernel methods (e.g., TTM). Therefore, this work focuses only on analytical insights that these numerical tools have not offered.
2. Our work further shows that obtaining the bath spectral function from reduced dynamics is possible. We are unaware of any work that achieves this based on QUAPI/SMatPI or TTM. Since the referee writes, “*they do facilitate the linking of open-system dynamics with the influence functional, enabling, for instance, the reconstruction of environmental spectral density from reduced dynamics,*” we would be happy to answer more if the referee could provide us with relevant references. To our knowledge, this mapping has never been shown in these approaches.
3. Our manuscript goes beyond the simplest spin-boson model set-up that most QUAPI/SMatPI works have focused on. We provide details for models with fermionic and spin baths, non-commuting system-bath coupling, and couplings with common baths. On the other hand, TTM could handle these cases as a general numerical tool, but it does not offer analytical insights. We show our analytical insights going beyond the simplest setup.

We have made appropriate edits in our manuscript to describe these main points more clearly. For instance, in the main text, we added a paragraph that starts with “*Overview of our contribution*” to summarize our contribution. After reorganizing our Appendix, we expanded our previously brief remarks on comparison with existing numerical approaches. This is in Appendix C C, and the corresponding text is as follows:

1. *QUAPI*: As explained in our main text, QUAPI assumes $I_k = 1$ for $k > k_{\max}$. This assumption alone does not immediately reveal U in terms of I_k . If one wrote U in terms of I_k using QUAPI without truncation, one would recover the same form of SMatPI without truncation (vide infra).

2. *SMatPI*: The SmatPI decomposition writes U in terms of influence functions and the bare system propagator. However, in doing so, it loses the time-translational symmetry of the memory functions, which makes the subsequent analysis we perform in this manuscript (see Appendix B and Appendix C F) inaccessible. Furthermore, the SMatPI approach was never formulated for problems in *Classes 2–4*.
3. *TTM*: The TTM decomposition, a black-box, data-driven, extrapolative technique, requires projection-free “numerical” inputs. In our analysis, we decompose the reduced system dynamical map in terms of the reduced system bare propagator and influence functions. This major insight is not present in TTM.

These differences make our analysis and insight presented in this manuscript unique contributions in this field.

We also refer readers to this explicit section in the main text with the following sentence:

Readers interested in the relationship between our work and existing numerical tools are referred to Appendix C C.

Referee Comment 2:

Specifically, I would first consider a simple – even exactly solvable – model to exemplify more explicitly the differences between the relations defining the distinct approaches and the insights that can be gained through them.

Response: We thank the referee for this suggestion. In our previous manuscript, we chose the simplest form of the spin-boson model because there are exact numerical results available for the parameter regimes we picked (i.e., we used free-pole HEOM). In the revised version, we studied models with different types of Gaussian baths (bosonic, fermionic, and spin) and different assumptions. We added a description of these different types of models to the main text:

1. *Class 1*: With only single α for all baths j (such cases are henceforth indicated by dropping the subscript α), $\{\hat{S}_j\}$ are all diagonalizable, and furthermore that $\{\hat{S}_j\}$ are all *simultaneously* diagonalizable. That is, all terms in $\{\hat{H}_{I,j}\}$ commute. The spin-boson model, other models in the same universality class, and Frenkel exciton models for photosynthetic systems belong to this class.
2. *Class 2*: No terms in $\{\hat{S}_j\}$ commute but each term in $\{\hat{S}_j\}$ is diagonalizable. Generalizing the models in *Class 1* to multiple nonadditive baths typically leads to this case. Such systems may arise when considering non-adiabatic dynamics of systems involving strong coupling of electronic degrees of freedom coupled to quantized photonic modes.

3. *Class 3:* There are common baths for some $\hat{H}_{I,j,\alpha}$ and $\{\hat{S}_{j,\alpha}\}$ may or may not commute. Examples of such baths arise when considering decoherence in models of coupled qubits.
4. *Class 4:* No terms in $\{\hat{S}_j\}$ commute and each term in $\{\hat{S}_j\}$ is not diagonalizable. The Anderson impurity model is representative of this category.

These cover simple and yet prototypical open-quantum system models in the literature, for which we offer insights.

Referee Comment 3:

Subsequently, I would utilize a more structured scenario to compare the complexity and efficiency of the methodologies.

Response: We thank the referee for this suggestion. Our previous manuscript included structured spectral density results in the Appendix and a pointer sentence in the main text. Our current manuscript retains the pointer sentence to the relevant part of the Appendix:

Furthermore, we show how this procedure can extract highly structured spectral densities as well in Appendix H and Fig. 12.

Response to Referee 2:

Referee Comment 0:

Ivander, Lindoy and Lee present a framework for connecting the generalised quantum master equation (GQME) to influence function methods. Their method is elegant, and as an example they show it is effective for simulating the spin-boson model. The paper is well written and the method very clearly explained with appropriate diagrams.

As the method presented seems closely related to other methods - which I will ask about further below - I am not sure whether this paper merits publication in Nature Communications. However, it is an interesting paper for specialists in the field - and the question is whether it represents a sufficiently important advance to fit within the journal's scope.

Response:

We thank the referee for reading our work and providing helpful feedback. We appreciate that the referee finds our work interesting.

Referee Comment 1:

1. My main question is: can the authors say how their approach is different from Makri's small matrix path integral approach, which in turn is similar to Cerrillo and Cao's transfer tensor method? There is some mention of this in appendix A, but could I ask the authors to expand on this? What does their technique enable that these other methods do not?

Response:

We thank the referee for their comment. As noted by the referee, in Appendix A of our original submission, we made brief remarks on the relationship of our approach with TTM and SMatPI (identical to QUAPI when untruncated). We used this opportunity to expand our discussion further. There are three points we would like to emphasize:

1. Our manuscript is not about developing efficient numerical simulation techniques for non-Markovian dynamics. Instead, we offer new analytical insights that connect two major paradigms in understanding non-Markovian dynamics: influence functional-based path integral (e.g., QUAPI, SMatPI) methods and memory kernel methods (e.g., TTM). Therefore, in this work, we focus only on analytical insights that these numerical tools have not offered.

2. Our work then further shows that it is possible to obtain the bath spectral function from reduced dynamics. We are unaware of any work that achieves this based on QUAPI/SMatPI or TTM.
3. Our manuscript goes beyond the simplest spin-boson model set-up that most QUAPI/SMatPI works have focused on. We provide details for models with fermionic and spin baths, non-commuting system-bath coupling, and couplings with common baths. On the other hand, TTM could handle these cases as a general numerical tool, but it does not offer analytical insights. We show our analytical insights go beyond the simplest setup.

We have made appropriate edits in our manuscript to describe these main points more clearly. For instance, in the main text, we added a paragraph that starts with “*Overview of our contribution*” to summarize our contribution. After reorganizing our Appendix, we expanded our previously brief remarks on comparison with existing numerical approaches. This is in Appendix C C, and the corresponding text is as follows:

1. *QUAPI*: As explained in our main text, QUAPI assumes $I_k = 1$ for $k > k_{\max}$. This assumption alone does not immediately reveal U in terms of I_k . If one wrote U in terms of I_k using QUAPI without truncation, one would recover the same form of SMatPI without truncation (vide infra).
2. *SMatPI*: The SMatPI decomposition writes U in terms of influence functions and the bare system propagator. However, in doing so, it loses the time-translational symmetry of the memory functions, which makes the subsequent analysis we perform in this manuscript (see Appendix B and Appendix C F) inaccessible. Furthermore, the SMatPI approach was never formulated for problems in *Classes 2–4*.
3. *TTM*: The TTM decomposition, a black-box, data-driven, extrapolative technique, requires projection-free “numerical” inputs. In our analysis, we decompose the reduced system dynamical map in terms of the reduced system bare propagator and influence functions. This major insight is not present in TTM.

These differences make our analysis and insight presented in this manuscript unique contributions in this field.

We also refer readers to this explicit section in the main text with the following sentence:

Readers interested in the relationship between our work and existing numerical tools are referred to Appendix C C.

Referee Comment 2:

When I read in the introduction to the paper that “efficient and completely general” descriptions of open quantum systems are yet to be found, I thought this paper would be presenting such a method. Do the authors believe this to be the case? If so, I think they would need to move beyond linear coupling to a harmonic bath.

Response: We thank the referee for this comment. We agree that the statement

Despite significant research efforts in this area, an efficient and completely general description of open quantum dynamics, especially those with memory (i.e., non-Markovianity), is yet to be found.

may imply that we have found such a method, which we never intended to say. We have, therefore, removed this statement.

Referee Comment 3:

Indeed, is it possible to use the technique, for example, for fermionic baths? If so, I would ask the authors to provide an example. If this is not the case, then I would ask the authors to reword the introduction to make this clear.

Response: We thank the referee for this comment. Yes, our technique can be applied to other types of Gaussian baths (fermionic and spin). In the revised version, we studied models with different types of Gaussian baths (bosonic, fermionic, and spin) and different assumptions. We added a description of these different types of models to the main text:

1. *Class 1:* With only single α for all baths j (such cases are henceforth indicated by dropping the subscript α), $\{\hat{S}_j\}$ are all diagonalizable, and furthermore that $\{\hat{S}_j\}$ are all *simultaneously* diagonalizable. That is, all terms in $\{\hat{H}_{I,j}\}$ commute. The spin-boson model, other models in the same universality class, and Frenkel exciton models for photosynthetic systems belong to this class.
2. *Class 2:* No terms in $\{\hat{S}_j\}$ commute but each term in $\{\hat{S}_j\}$ is diagonalizable. Generalizing the models in *Class 1* to multiple nonadditive baths typically leads to this case. Such systems may arise when considering non-adiabatic dynamics of systems involving strong coupling of electronic degrees of freedom coupled to quantized photonic modes.
3. *Class 3:* There are common baths for some $\hat{H}_{I,j,\alpha}$ and $\{\hat{S}_{j,\alpha}\}$ may or may not commute. Examples of such baths arise when considering decoherence in models of coupled qubits.
4. *Class 4:* No terms in $\{\hat{S}_j\}$ commute and each term in $\{\hat{S}_j\}$ is not diagonalizable. The Anderson impurity model is representative of this category.

These cover simple and yet prototypical open-quantum system models in the literature, for which we offer insights.

For fermionic baths that the referee asked about, we considered three cases in the revised manuscript:

1. A specific realization of the spin-fermion model that can be bosonized into the spin-boson model (Class 1). See Appendix C B 2 for more details.
2. A spin coupled to a Gaussian fermionic model with a diagonalizable coupling (Class 1). See Appendix C B 3 for more details.
3. A fermionic impurity coupled to a fermionic bath with a non-diagonalizable coupling (Class 4). This corresponds to the single impurity Anderson model. See Appendix F for more details.

While other than Class 1, we do not present a general diagrammatic algorithm to construct the memory kernel (\mathbf{K}) from influence functions (\mathbf{I}), in all cases, we show how to relate \mathbf{K} and \mathbf{I} analytically and how to obtain the bath spectral density from reduced dynamics.

Referee Comment 4:

In my view the most important part of this paper is the observation that one can reconstruct the spectral density $J(\omega)$ from the time evolution of ρ - and this is nicely illustrated with a couple of examples. How general is this observation? Does it only apply for linearly coupled harmonic baths or more generally?

Response: Thank you for this encouraging comment on the importance of our contribution. Our approach assumes Gaussian baths and bilinear system-bath coupling. In our revised manuscript, we present how to reconstruct the bath spectral density for the four *Classes* we mentioned in our response to Comment 3. The models we considered include bosonic, fermionic, and spin baths. Our contribution is also clearly stated in the main text in the paragraph that starts with “*Overview of our contribution*”.

Referee Comment 5:

At the bottom of col 1, page 4. Can the authors comment on the connecting between the T tensor and the process tensor as introduced in ref 31 of the paper? Also connect the PT to Eq. A71 and A72.

Response: Thank you for bringing the process tensor formalism to our attention. The process tensor framework is a formal and general methodology to characterize non-Markovian quantum dynamics (Phys. Rev. Lett. 123, 240602 (2019)). Indeed, it is known in the literature that the

process tensor (PT) is the generalization of influence functional (IF) in the sense that both objects encode the influence of the environment on the dynamics of the reduced system. Furthermore, its form is known in terms of the influence functions (\mathbf{I}), and we assume that this motivated the referee to wonder if our \mathbf{T} tensor is identical to the PT.

The central difference between PT and our \mathbf{T} tensor is as follows. The PT is a tensor used to obtain the system dynamical map, \mathbf{U} . In essence, one can contract the PT with appropriate products of the bare system propagator to obtain \mathbf{U} . Going from \mathbf{U} to \mathbf{K} requires a non-trivial re-arrangement of the PT terms (e.g., the Dyck path). Because of this, there seems to be no direct relationship between PT and the memory kernel, \mathbf{K} . Our \mathbf{T} tensor represents a PT-like tensor for the memory kernel, \mathbf{K} . When contracted with appropriate bare system propagators, \mathbf{T} can directly yield \mathbf{K} . The same relationship has been elusive for the PT; this contribution is our unique one.

We have added the following sentences in the manuscript to clarify the connection between the PT and the \mathbf{T} tensor, as pointed out by the referee:

1. In the main text:

The $T_{N;\bullet}$ tensor appears to be related to the process tensor. \mathbf{T} represents \mathbf{K} upon the contraction with \mathbf{P} , but the process tensor is used to construct \mathbf{U} when contracted with \mathbf{P} . Subsequently, there is a non-trivial rearrangement of the terms to write \mathbf{K} in terms of the process tensor. The simple relationship between \mathbf{T} and \mathbf{K} in Eq. (16) is our unique contribution.

2. In Appendix C H:

As mentioned in the main text, \mathbf{T} takes a similar form as that of the process tensor, but it is distinct in that it directly represents \mathbf{K} , as opposed to \mathbf{U} as in the process tensor literature.

Referee Comment 6:

Fig 3, for the spin boson model, can the method capture the phase transition in this model?

Response: Thank you for this question. Reiterating our response to Comment #1, we emphasize that our work is not about developing efficient numerical simulation methods. Therefore, we assumed that the referee wanted to know if the diagrammatic construction of the memory kernel is still valid for the regimes near the phase transition (i.e., the regimes with an algebraic decay of memory in time.) Our approach is exact and non-perturbative, so it is, in principle, valid in any regimes of the spin-boson model.

In our original manuscript, we analyzed a test case close to the localization transition in the sub-ohmic regime, shown in Fig 2c (for the norm of the memory kernel) and Fig 3c1. Fig 2c shows clearly that the memory of the system dynamics decays very slowly in that regime (as indicated by the norm of \mathcal{K}). Such a slow decay is captured by our decomposition techniques. Since this is an interesting point to mention in the main text, we added the following sentences:

We note that describing quantum phase transitions at $T = 0$ would require capturing the algebraic decay in \mathbf{I}_N . Our analysis can, in principle, capture such a slow decay as our approach is exact but will require further optimization in the underlying numerical algorithms for practical applications.

Referee Comment 7:

More generally, how do computation times/resources using this method compare to other techniques (like TEMPO)?

Response: We thank the referee for this question. In line with our response to Comment #1, the main results of this work are best interpreted as novel analytic insights. Although ample opportunities exist in using and developing our diagrammatic construction as a numerical approach to approximate open quantum system dynamics (e.g., by considering certain dominant diagrams), these are beyond this work's scope. Likewise, numerical heuristics could be developed to enable efficient reconstruction of the bath spectral density from reduced system dynamics, a task that was elusive in this field before our work.

To highlight this opportunity, we have the following sentences in the main text:

We believe that this unified framework for studying non-Markovian dynamics will facilitate the development of new analytical and numerical methods that combine the strengths of both GQME and INFPI. For example, deep connections between the present work and recent matrix product state (MPS)-based approaches invite ideas that would efficiently extract the environmental spectral density from reduced system dynamics.

Referee Comment 8:

Eq. A28. Is there a K here that should be an F?

Response: We thank the referee for pointing out a typo. This has been corrected.

Referee Comment 9:

Can the authors clarify the comment below Eq. A60 about frequencies that make the denominator go to zero? I could not quite follow the comment about these not contributing to $F(\omega)$, sorry! I can see that $\delta k > 0$ are H.c. of $\delta k < 0$ but does this not leave a real part?

Response: We thank the referee for this question and for carefully reading our manuscript. We apologize for the confusion this has caused. The referee is correct, and we have referenced an incorrect equation in this statement. As pointed, looking at (now) Eq. (A85) these nodal frequencies will **still** contribute to $\mathcal{F}(\omega)$ as the real part of $\eta_{\Delta k}$ will not vanish. Therefore, Eq. (A85) alone will not imply that at these nodal frequencies, the spectral density does not affect the reduced dynamics of the reduced system. Only when we refer to (now) Eq. (A83) can we see that the real parts are indeed also identically zero.

We have now corrected this statement, rephrased it for clarity, and we have added a few intermediary steps:

One must pay special attention to frequencies with $\omega\Delta t/2 = n\pi$ for integer n as the denominator in Eq. (A86) is zero and the expression diverges. However, this is not a problem because, physically, the spectral density at such nodal ω values does not alter the reduced system dynamics. This is because they do not contribute to $\mathcal{F}(\omega)$ as shown in Eq. (A83), and can thus be seen as the null kernel of the map $J(\omega) \rightarrow \eta$. To see this, we substitute these frequencies to Eq. (A85)

$$\begin{aligned}
\mathcal{F}\left(\frac{2n\pi}{\Delta t}\right) &= \frac{\Delta t}{2\pi} \sum_{\Delta k=-\Delta k_{\max}}^{\Delta k_{\max}} \eta_{\Delta k} e^{i2n\pi\Delta k} \\
&= \frac{\Delta t}{2\pi} \sum_{\Delta k=-\Delta k_{\max}}^{\Delta k_{\max}} \eta_{\Delta k} \underbrace{\cos 2n\pi\Delta k + i \sin 2n\pi\Delta k}_1 \\
&= \frac{\Delta t}{\pi} \sum_{\Delta k=0}^{\Delta k_{\max}} \text{Re} \eta_{\Delta k}. \tag{1}
\end{aligned}$$

However, as we substitute the nodal frequencies to Eq. (A83) we then see that indeed the real part of $F\left(\frac{2n\pi}{\Delta t}\right)$ also vanish identically:

$$\mathcal{F}\left(\frac{2n\pi}{\Delta t}\right) = \frac{2}{\pi} \frac{J(\omega)}{\omega^2} \frac{\exp\{\beta\hbar\omega/2\}}{\sinh \beta\hbar\omega/2} \underbrace{\sin^2 2n\pi}_0 = 0. \tag{2}$$

Referee Comment 10:

In a comment in the caption of Fig 11, the author say that going from $K \rightarrow I$ for high orders is infeasible. Why? Does this limit the applicability of the method?

Response: We thank the referee for this comment. Indeed, in calculating I_N from \mathcal{K}_N with our present approach, we must compute explicitly Catalan's numbers $C_N = \frac{1}{N+1} \binom{2N}{N}$ of diagrams which ultimately make up \mathcal{K}_N , therefore the costs to compute higher order I is exponential in N . Furthermore, for each term, we must perform $N - 1$ contractions. The exact evaluation of the memory

kernel shown by our work is limited due to this exponential scaling bottleneck. However, there are still opportunities to numerically approximate the memory kernel (beyond the scope of the current manuscript), for example, by including specific classes of Dyck diagrams (see Appendix C E). We have now clarified this point by adding the following comments to the manuscript:

1. In the paragraph below Eq. (A78)

We note the computational costs of procedure therefore scale (super)combinatorially with the Dyck order N as one must compute Catalan's numbers $C_N = \frac{1}{N+1} \binom{2N}{N}$ of diagrams in \mathcal{K}_N , each with about $N - 1$ of contractions. However, there are ample opportunities to improve this procedure numerically, for example by including specific classes of Dyck diagram (see Appendix C E).

2. In the caption of Fig (12)

(see paragraph below Eq. (A78))

Referee Comment 11:

A further question: how well can one extract U from experimental measurements?

How many such measurements need to be done to do this to a particular accuracy?

Response: We thank the referee for these questions. It is difficult for us to speculate how well one can extract U from experimental measurements because the accuracy of measurements depends on many different factors. Assuming an additive error, ϵ , in $\rho(t)$ and a perfect implementation of the initial $\rho(0)$, the error bound for U is also ϵ . Typical quantum devices then would have $\epsilon \propto 1/\sqrt{N_{\text{measure}}}$ dependence, which is a good estimate for the number of measurements for a desired accuracy. We are currently in the process of designing such an experiment and are quite optimistic about the prospects of demonstrating the bath spectral density reconstruction in a real experiment.

Response to Referee 3:

Referee Comment 0:

The article discusses the dynamics of an open quantum system, described either using the Nakajima-Zwanzig memory kernel or the Feynman-Vernon influence functional path integral. The main result is a formal, diagrammatic connection between the two approaches, which allows for the expression of the memory kernel through the influence functions (or vice versa), using the reduced system evolution operator to link the two methods. As an application of this connection, the authors show how to construct the bath spectral density using only the reduced dynamics as input.

Altogether, I find that the results are interesting and clearly contribute to the field. I therefore recommend publication. However, I have some minor remarks.

Response:

We thank the referee for reading our work, for their encouraging remarks, and for recommending for publication. In regards to the referee's minor comments we have enumerated our responses in what follows.

Referee Comment 1:

The authors mention that their construction of K using I is non-perturbative. This is not clear to me. In the context of GQME approaches one often expands the memory kernel K in orders of the system-environment coupling. When this is done in time-space one finds that higher-order memory kernel terms contribute "later in time" than lower order ones (in the sense of first non-vanishing Taylor coefficients). In this sense, e.g. K_3 from the paper contains higher order contributions than K_2 , which seems to give a perturbative character to the set of equations (7)-(10).

Response: We thank the referee for this comment. The recursive structure of the memory kernel is not the manifestation of a perturbative character. In our construction, there is no small parameter that is used for perturbative expansions. Eqns (7)-(10) are **exact** up to the Trotter error inherent in discretized path integral-based methods. These expressions are valid for any coupling strength in the models considered. We added a clarifying comment in the main text:

We emphasize that Eqs. (7) to (10) are exact up to the Trotter discretization error and valid for any coupling strengths in the models considered in this work. By definition, earlier \mathcal{K}_N contains shorter memory effects and will thus appear simpler.

Referee Comment 2:

The article is concerned with systems with bosonic environments. However, both the Feynman-Vernon path integral approach and the Nakajima-Zwanzig approach are also commonly used to also describe fermionic environments, e.g. in the description of quantum transport. Are there principle limitations in finding a similar formal connection between I and K also in the fermionic case?

Response: We thank the referee for this comment. Our technique can be applied to other types of Gaussian baths (fermionic and spin). In the revised version, we studied models with different types of Gaussian baths (bosonic, fermionic, and spin) and different assumptions. We added a description of these different types of models to the main text:

1. *Class 1:* With only single α for all baths j (such cases are henceforth indicated by dropping the subscript α), $\{\hat{S}_j\}$ are all diagonalizable, and furthermore that $\{\hat{S}_j\}$ are all *simultaneously* diagonalizable. That is, all terms in $\{\hat{H}_{I,j}\}$ commute. The spin-boson model, other models in the same universality class, and Frenkel exciton models for photosynthetic systems belong to this class.
2. *Class 2:* No terms in $\{\hat{S}_j\}$ commute but each term in $\{\hat{S}_j\}$ is diagonalizable. Generalizing the models in *Class 1* to multiple nonadditive baths typically leads to this case. Such systems may arise when considering non-adiabatic dynamics of systems involving strong coupling of electronic degrees of freedom coupled to quantized photonic modes.
3. *Class 3:* There are common baths for some $\hat{H}_{I,j,\alpha}$ and $\{\hat{S}_{j,\alpha}\}$ may or may not commute. Examples of such baths arise when considering decoherence in models of coupled qubits.
4. *Class 4:* No terms in $\{\hat{S}_j\}$ commute and each term in $\{\hat{S}_j\}$ is not diagonalizable. The Anderson impurity model is representative of this category.

These cover simple and yet prototypical open-quantum system models in the literature, for which we offer insights.

For fermionic baths that the referee asked about, we considered three cases in the revised manuscript:

1. A specific realization of the spin-fermion model that can be bosonized into the spin-boson model (Class 1). See Appendix C B 2 for more details.
2. A spin coupled to a Gaussian fermionic model with a diagonalizable coupling (Class 1). See Appendix C B 3 for more details.

3. A fermionic impurity coupled to a fermionic bath with a non-diagonalizable coupling (Class 4). This corresponds to the single impurity Anderson model. See Appendix F for more details.

While other than Class 1, we do not present a general diagrammatic algorithm to construct the memory kernel (\mathbf{K}) from influence functions (\mathbf{I}), in all cases, we show how to relate \mathbf{K} and \mathbf{I} analytically and how to obtain the bath spectral density from reduced dynamics.

REVIEWERS' COMMENTS

Reviewer #1 (Remarks to the Author):

I thank the authors for their reply and I acknowledge that the newly added investigation of different types of Gaussian models certainly improves the quality of the manuscript.

However, unfortunately, I think that the main concerns I had in my previous report have not been addressed. Thus, I do not recommend the manuscript for publication in Nature Communications (but I would suggest the author to submit it to a more specialized journal).

My main points are the following ones.

- I understand that the manuscript is not about “developing efficient numerical techniques” but rather on a conceptual point – i.e., the analytical connection between the influence functional and the memory-kernel master equation. However, my remark concerns the potential impact of the result: while for researchers working on/with open quantum systems this connection is certainly of interest, by itself it does not guarantee the broad impact necessary for a high-impact journal. The physical insights provided by this new result are certainly significant, but their relevance compared to existing analytical and numerical methods is quite subjective (see, for example, the third point below); this is why an extended comparison with existing methods, specifically from the point of view of which models can be (efficiently) described by them, would be necessary for publication in Nature Communications.

- Several approaches to open quantum systems put forward strategies to obtain the spectral function from the reduced dynamics. Besides the well-established field of noise spectroscopy mentioned by the authors (see Phys. Rev. B 93, 121407 (2016) for a treatment beyond standard pure dephasing), one can think to the most recent machine-learning based strategies, J Barr et al 2024 Mach. Learn.: Sci. Technol. 5 015043 (2024). Although these methods might not be directly based on QUPAI techniques, I reiterate my previous point: the manuscript presents a new method to obtain the spectral density from the reduced dynamics, but without an extended comparison with other approaches it is not possible to assess the impact of such a method and then to justify its publication in Nature Communications.

- I also disagree with the statement that TTM does not provide analytical insights. As shown in F.A. Pollock, and K. Modi, Quantum 2, 76 (2018), transfer tensors (TT) can indeed be

linked to the microscopic details of the system-environment interaction—see equations (4)-(7) therein. For instance, their structure can be related to the environmental state, the system-environment correlations and thus ultimately to the memory time characterizing the model at hand. I am not claiming that this encompasses the physical insight provided by the authors' approach, but my point is that also existing methods provide general analytical insights, so that this alone does not justify the uniqueness of the model introduced here.

Reviewer #2 (Remarks to the Author):

Ivander, Lindoy and Lee have done an extremely thorough job of responding to my previous comments. In particular, they now extend their analysis to four different classes of interaction Hamiltonians and include fermionic baths. They have also clarified how their analysis differs from other techniques such as SMatPI, and also explained how the method put forward here is different to the process tensor.

I have decided, on balance that the work will be of significant interest to specialists in the field, and so this paper can be published in Nature Communications.

Reviewer #3 (Remarks to the Author):

I thank the authors for addressing my comments and recommend publication.